# Strong Ground Motion Sensor Network for Civil Protection Rapid Decision Support Systems

**DOI:** 10.3390/s21082833

**Published:** 2021-04-17

**Authors:** Georgios Chatzopoulos, Ilias Papadopoulos, Filippos Vallianatos, John P. Makris, Maria Kouli

**Affiliations:** 1Institute of Physics of the Earth’s Interior and Geohazards, UNESCO Chair on Solid Earth Physics and Geohazards Risk Reduction, Hellenic Mediterranean University Research Center, 73133 Crete, Greece; gechat@hmu.gr (G.C.); ilias@uwiseismic.com (I.P.); fvallian@geol.uoa.gr (F.V.); mkouli@hmu.gr (M.K.); 2Seismic Research Centre, University of West Indies, St. Augustine, Trinidad and Tobago; 3Section of Geophysics–Geothermics, Department of Geology and Geoenvironment, National and Kapodistrian University of Athens, Panepistimiopolis, 15784 Athens, Greece

**Keywords:** strong ground motion, sensor network, decision support system, spectral acceleration, arias intensity, PGA, PGV, PGD, GIS

## Abstract

Strong motion sensor networks deployed in metropolitan areas are able to provide valuable information for civil protection Decision Support Systems (DSSs) aiming to mitigate seismic risk and earthquake social-economic impact. To this direction, such a network is installed and real-time operated in Chania (Crete Island, Greece), city located in the vicinity of the seismically active south front of the Hellenic Subduction Zone. A blend of both traditional and advanced analysis techniques and interpretation methods of strong ground motion data are presented, studying indicative cases of Chania shaking due to earthquakes in the last couple years. The orientation independent spectral acceleration as well as the spatial distribution of the strong ground motion parameters such as the Peak Ground Acceleration (PGA), Peak Ground Velocity (PGV), Peak Ground Displacement (PGD) and Arias Ιntensity observed at the urban area of Chania are presented with the use of a Geographic Information System (GIS) environment. The results point to the importance of the strong ground motion networks as they can provide valuable information on earthquake hazards prior to and after detrimental seismic events to feed rapid systems supporting civil protection decisions for prevention and emergency response.

## 1. Introduction

Strong earthquakes are infrequent events causing in many cases enormous losses, costs, and devastation, especially when their epicenter is close to residential areas. Nevertheless, similar consequences may also result from moderate yet adjacent seismic activity, especially at regions of high seismic risk. Urbanization increases worldwide, not excluding earthquake prone areas. Many metropolitan cities are located in heterogeneous subsoils, in a complex geologic and geotectonic setting, and in a lot of cases on alluvial deposits to exploit river or sea roads of commerce and culture. The expansion of urban areas and their surroundings very often leads to inappropriate land use and more vulnerable structures and buildings.

Cioca and Cioca [1] conceptualize the fundamental definitions involved: (Natural) Hazard is the probability that a potentially destructive natural phenomenon, like an earthquake, might occur and impinges upon human lives and their social-economic environment by breaching existing safety limits and civil protection. Vulnerability denotes the susceptibility to injury or loss, damage, or destruction, while exposure refers to beings, premises, infrastructure, and/or the environment in relation to a specific natural hazard. Risk is defined as the expected death toll and level of injuries, and the impeding impact on properties, structures, and networks due to isolated or combined hazards, constrained in space and time.

In this context, earthquake hazard mitigation and seismic risk reduction to metropolitan areas is a complex, multidisciplinary, spatiotemporally dynamic problem, difficult to formulate. Thus, it requires extensive and continuous studies to understand the physical mechanisms as well as the catastrophic implications of earthquakes.

As populations increase and economies grow in big or mega urban areas, an augmented seismic risk is expected, posing accessional pressure to civil protection decision-makers to abate this risk. Thus, the development and use of Decision Support Systems (DSSs) for Natural Hazard Risk Reduction (NHRR) gains significant importance [2].

A contemporary DSS is a mixture of high-end computer systems and data warehouses supporting efficient management of large volumes of data and complex information. Its fundamental scope is to improve and simplify decision processes and to stimulate the effectiveness of decisions to be made in all the hierarchical levels of decision-makers, belonging to diverse operators (organizations, agencies, authorities), end-users, and stakeholders [1].

DSSs for NHRR are classified as model-oriented DSSs that adopt model integration from a palette of simulation models for risk analysis (hazard, exposure, vulnerability modelling), optimization models for test and assess risk-reduction options, as well as suggestion models which provide suggested decisions for risk-reduction measures and emergency response, eliminating human bias. Nowadays, DSSs for NHRR take advantage of web-technologies, artificial intelligence, data mining, real-time, parallel processing, Geographic Information System (GIS), remote sensing, mobile and satellite communications.

It is prevalent that proactive seismic risk reduction is preferable and much more effective, in all aspects, compared to civil protection actions for response and recovery. Inescapable outcomes should be the public dissemination of information on seismic risk and eventually seismic early warning. Hence, seismic-risk reduction planning and prevention strategies should be short-, intermediate- and long-term monitoring and assessing risk variations [3,4,5,6]. To meet the ultimate objectives, seismic risk criteria are determined and examined. Newman et al. [2] recapitulate the different types: economic (business interruption and productivity losses, damage of properties, critical infrastructure, lifelines and pipelines, public buildings, schools, hospitals; transport, energy, and communication networks), social (fatalities, casualties, people requiring short- or long-term post-hazard assistance, unequal hazard impact on different ethnic, demographic or regional populations, social distress, evacuation upheaval, homelessness and displacement, loss of public services), environmental (terrestrial, aquatic, protected areas), and other (political implications, loss of credibility of local authorities, safety, cultural heritage, universal monuments). Furthermore, a DSS for seismic risk reduction must incorporate risk-assessment methodology to monitor and evaluate the effectiveness of different risk-reduction measures undertaken, using decision indicators such as direct losses, indirect costs, risk-reduction implementation costs, possible indirect benefits from earthquake hazard, negative and/or positive side-effects.

Earthquake hazard zonation for urban areas (seismic microzonation) is the primary step toward a seismic risk analysis and reduction strategy. Seismic Hazard Analysis (SHA) is a multidisciplinary modeling that aims to forecast earthquake occurrence and its resultant ground shaking [7]. The SHA aim to evaluate the ground intensities with a probabilistic (PSHA) or a deterministic (DSHA) approach. The first one relays on recurrence relation, like the Guttenberg–Richer law, while the second one focuses on the structures design based on Peak Ground Acceleration (PGA) and spectral acceleration values [8]. A modern approach, the neodeterministic (NDSHA) is a scenario-based procedure that takes earthquake source, seismic wave propagation path and local site effects into account, to supply realistic time histories of strong ground motions [9].

Well-instrumented urban areas are able to produce a large database of strong-motion recordings. These results can improve the empirical characterization of ground motion and help to better understand the seismicity patterns. We present the deployment of a Strong Ground Motion Network (SGMN) in a metropolitan area (Chania, Crete Island, Greece) together with multiparametric analysis techniques of the data collected in recent indicative cases of earthquake occurrence. The dense SGMN is a permanent installation providing in near real-time hazard information for the ground motion that can automatically feed a DSS for seismic risk reduction and emergency response. Compared to the traditional microzonation techniques which usually rely on single-site measurement or temporal seismometer deployments, the SGMN creates continuously a detailed image of the ground intensity parameters for an area with difficult geological setting and helps to study the various effects of the seismic waves.

## 2. Research Area, Sensors, and Data

The monitoring area consists of a large basin with sediments that vary in thickness and stiffness which is ideal for deploying a dense strong ground motion network with sensors installed on different geologic formations to record the ground shaking variations. Crete is located in the southern part of Greece and depicts diversified geological setting (Figure 1a). As described by [10,11], the whole island is a complex structure with continuous nappe units such as Paleozoic to Triassic Phyllite-Quartzite (brown color in Figure 1a) along with the lowest formations comprised by Jurassic white-grey recrystallized limestones and marbles with considerable thickness (blue color in Figure 1a) along with Permian schist type rocks and shallow sea carbonates (pink color in Figure 1a). Superincumbent the bedrock formation, are the large sedimentary basins with Neogene formations, mainly consisted of marles, sandstones (bronze color in Figure 1a) and limestones (beige color in Figure 1a) with Terra rossa weathering (magenta color in Figure 1a). To the extent of the area under study, they are visible in the northern and eastern part of the city, as well as in the neighboring Akrotiri peninsula and at the foot of the mountain range. The Quaternary deposits (white color in Figure 1a) are loose to very loose material, such as sands, clays and gravels with considerable thickness that is estimated about 150 m [12], covering the bigger part of the southern Chania suburban area. In Figure 1a, the yellow lines depict possible active faults while the green ones are inactive. The prevalent geological formations in each SGM-station location and the sensor type installed are summarized in Table 1.

SGMN utilizes two-generation sensors of the 24 bit integrated tri-axial accelerometer produced by REF TEK [13]. The first type, the more recent one, is the 130-SM, equipped with a 131-8019 sensor that works on a full-scale range > ±3.5 g with ±10 V full scale voltage, while the second one is the ANSS/02 that has a 0131-02/03/I sensor which works with ±6.9 V full scale voltage and has ±3.5 g clip level. Both models are equipped with three microelectromechanical systems (MEMS), one for vertical and two for the horizontal components. Si-Flex MEMS have silicon proof mass and springs, specifications derived from the REF TEK sensor manuals [13]. A switched capacitor integrated circuit is used to calculate the mass position changes and a compensation circuit is used to center the mass with electrostatic forces. The MEMS accelerometers have no need for calibration except a periodically dc offset remove. Maintenance options, performed from time-to-time, are: firmware upgrade, replacement of internal battery or flash card, and check of the connectors. Four boards are assigned for different modules such as A/D conversion, signal processing, in-situ data storage on memory cards, and data transfer through internet telemetry.

All sensors are telemetrically connected via ADSL lines to a central server running the REF TEK Protocol Daemon (RTPD) software which provides remote control of the SGM-stations, data error-correction, archiving and storage. As the SGMN is part of the HC network, the strong ground motion recordings with 250Hz sample rate are stored in REF TEK PASSCAL archives [14] and in miniSEED format [15] at the Hellenic Seismological Network Crete (HSNC) data storage units [16,17,18,19]. In the framework of HELPOS (Hellenic Plate Observing System, Greece) action that aims to homogenize and unify the seismic networks in Greece, all the waveforms are stored and are available for distribution through the National European Integrated Data Archive (EIDA) node (http://eida.gein.noa.gr/) (accessed on 2 February 2021). The earthquake parameters i.e., magnitude, epicenter, hypocenter and origin time, in this work, accrue from the manual revised solution announced by the National Observatory of Athens seismic network (NOA) [20].

The investigating time period for the present work is from 2017 up to the first days of February 2021. All the earthquakes that recorded from the available strong ground motion sensors during the aforementioned time period have been examined and the events that gave rise to pronounced acceleration values are presented as study cases (Table 2). Among these examples are the distant, shallow, strong earthquake with magnitude M_L_ = 6.7 that occurred in 2020/10/30 offshore of Samos Island and the intermediate-depth, strong earthquake, M_L_ = 6.1, that occurred in 2019/11/27 with epicenter just a few tens of kilometers west of Chania. Smaller magnitude events that generated similar spectral acceleration values, due to their short epicentral distance from the urban area of Chania city, are also presented for fruitful comparison reasons.

## 3. Processing Methods

The site seismic response spectra are a way for the earthquake engineers to understand and study the behavior of structures during earthquakes. They represent the change of spectral acceleration in different periods. To calculate the response spectra, we adopted Boore’s method [21] that has the advantage of not relying on the orientation of the horizontal components. A single rotation of the recorded motions produces one time series of acceleration which is preferable for dynamic structure analysis [21]. According to this method, the largest absolute amplitude of each of the two horizontal component oscillator time series $\varepsilon_{1},\varepsilon_{2}$ with φ rotation angle is calculated by the equations:(1)$$\varepsilon_{1}\left(t,\varphi \right)=\varepsilon_{1}\left(t,0\right)cos\left(\varphi \right)+\varepsilon_{2}\left(t,0\right)sin\left(\varphi \right),$$
(2)$$\varepsilon_{2}\left(t,\varphi \right)=-\varepsilon_{1}\left(t,0\right)sin\left(\varphi \right)+\varepsilon_{2}\left(t,0\right)cos\left(\varphi \right),$$

The procedure to calculate the ground motion intensity involves the non-overlap rotation of horizontal axis to find the geometric mean $Geo_{m}$ of the period dependent response spectra $R_{sp}$:(3)$$Geo_{m}=\sqrt{R_{sp}\left(\varepsilon_{1}\right)R_{sp}\left(\varepsilon_{2}\right)},$$

Then, to calculate the period T independent response spectra $R_{sp}I$, a penalty function [14] is minimized based on the rotation angle φ:(4)$$R_{sp}I\left(T\right)=Geo_{m}\left(\varphi_{min},T\right),$$

This procedure requires time series with the instrument response removed. In the initial REF TEK files, the seismic signal was cut few seconds (8–15 s depends on the existence of transient noise) before P waves arrival up to the coda and then the files were transformed into SAC format. The SAC software and the EVALRESP code [22] have been used to prepare the files. A typical signal processing procedure followed for removing trend and offset, if any, and then the waveforms were tapered to avoid the presence of jumps at the time series borders. Finally, to use the Boore’s code [23], the SAC files turned into SMC format, which is the standard time series ASCII type format for strong motion data developed by U.S. Geological Survey [24]. The rotation step angle was set to one degree and the damping to 5% for the full set of periods. The resampling interpolation factor for the FFT was set to 8. Among the results obtained from the software, the median values (50% fractiles) of response spectra for the period-independent rotation angle were adopted and plotted against the period because these circumvent the very small geometric means which are associated to strongly correlated motions [21].

The typical measurements in a strong ground motion analysis were carried out, i.e., the peak ground acceleration (PGA), the peak ground velocity (PGV), the peak ground displacement (PGD). Another measurement provided, which is not based on a maximum value, the Arias Intensity $I_{A}$, the time-integral of the square of the ground acceleration, a(t), measures the cumulative energy per unit weight for the total duration Td of the ground motion [25,26]:(5)$$I_{A}=\frac{\pi }{2g}\overset{T_{d}}{\underset{0}{\int }}a{\left(t\right)}^{2}dt,$$

To present the spatial distribution of the ground shaking recorded from the SGMN and complete the data gaps between the measurements, different maps were created with an interpolation method in an ArcGIS environment [27]. In addition, to investigate the ground properties and feed with accessional information the Kriging interpolator, the single station site characterization method, proposed by Nakamura [28], was applied. Several passive source surveys have been carried out by means of a portable seismometer at all station sites to calculate the Horizontal vs Vertical Spectral Ratio (HVSR) of ambient noise. Although, Papadopoulos et al. [12] had conducted extensive HVSR study into a considerable part of the area of interest, we carried out new measurements at the exact sites of the strong ground motion sensors. The software Geopsy [29] has been used to process the waveforms and obtain the predominant frequency, *f*_0_, and amplitude, *A*_0_, from the ambient noise recordings (see Table 3). A bandpass filter with corner frequencies 0.2 Hz and 20 Hz respectively, and a mean value base correction have been used. The one-hour waveforms were separated in no overlapping, 60 s windows cosine tapered with 5% width at both ends and the smoothing constant was set to 40 [30]. The predominant frequency and amplitude values have been used as covariates for the ordinary Co-Kriging interpolation method. The ordinary Co-Kriging equation to estimate a missing value is:(6)$$Z\left(s_{0}\right)=\overset{n}{\underset{i}{\sum }}\left(\lambda_{i}z_{1}\left(s_{i}\right)+\mu_{i}z_{2}\left(s_{i}\right)\right),$$
where the n,i are the number of neighbors and the search points, λ,μ are the weights of the variables and $z_{1},z_{2}$ the measured values at location s [27]. The Co-Kriging is the well-known geostatistical interpolation method with the addition that it incorporates one or several well-sampled variables so it can be a more effective estimator [31]. Among the different models that affect the range, sill, and nuggets, we selected the one that came out with the lowest errors between predicted and measured values. Focus has been paid to the measurement points to keep the ground intensity values unchanged as possible from the interpolation procedure. Other options to obtain the better interpolation images were to set the minimum and maximum sampling neighbors to six and eleven and the area around a point was divided into eight search sectors to smooth the results [19].

## 4. Results

The installation of the SGMN’s sensors on different geological formations to cover the broad area of a metropolitan city, the Chania in our case, provides a near real-time realistic delineation of the ground shaking in the monitoring area. The response spectra from all stations after a major seismic event present valuable insight earthquake aftermath in different parts of the city which can be used by civil protection agencies, local authorities, and other stakeholders.

Figure 2 illustrates the spectral acceleration plots in g-units for the events that brought on the largest values. The majority of buildings in the research area are of low structure height, typically exhibit natural periods much less than 1 s, thus all the panels on the Figure 2 are limited to periods up to 1 s. An exception to that is the Samos event panel (e) which depicts in the response spectra noticeable acceleration values for periods up to 2 s possibly related to the weak attenuation of low frequency part of spectrum with distance. The EQ.3, intermediate depth, event along with the EQ.4 event with epicentral distance less than 40 km from Chania were the two strongest recordings of the SGM network during the examination period. The spectral acceleration values are slightly above the 9% g for one station while the rest stations in alluvial deposits reported values more than 5% g. A similar behavior with slightly reduced acceleration exhibited the EQ.1 minor event with epicenter located southwest of Chania at very short distance from the city. The EQ.5 event had similar parameters (hypocenter and magnitude) with the EQ.4 one, but it presented considerably smaller acceleration values as the maximum was close to 4% g. The Samos strong event occurred very far away (about 365 km) from the Chania metropolitan area, but its magnitude was large enough to record almost 4% g at two strong ground motion stations. The EQ.8 is the most recent event felt in Chania in the same direction with the 2017′s one (EQ.1), similar depth and larger magnitude but the factor of greater distance is the reason for the much smaller acceleration values which were less than 2% g.

For the case of Chania metropolitan city, where the sedimentary basin formations consist of materials with different stiffness and thickness from site to site, it is very interesting to examine the response spectra in each station for a number of strong shocks, as this can highlight the important footprint of local site effects. All events with significant ground shaking intensity since 2017, recorded from the majority of stations were examined. The spectral acceleration plot results point out that there are considerable variations in the reported values among some locations. From the results of this investigation, we extract and present in Figure 3 the three stations (G0LD, MULT, and SOUD) that systematically reported higher acceleration values, due to local subsurface geology, as opposed to the three stations (ATEI, LENT, and POLY) that consistently recorded lower values. The stations located in the Quaternary deposits demonstrated in most cases a clear peak between the periods 0.1 and 0.3 s. The EQ.3 and EQ.4 events generated significantly larger acceleration values at these stations. On the contrary, the stations on the Neogene limestones do not reveal any sharp peak, most events have reported values less than 2% g while the maximum acceleration, attributed to the M_L_ 6.1 event near Chania barely, reaches the value of 4% g.

The dense SGM network is capable of originating multi-informal ground motion parameter distribution maps in a very rapid and direct way after a large shaking. The PGA in cm/s^2^, PGV in cm/s, PGD in mm and the Arias intensity in cm/s maps for the two recent earthquakes with the largest acceleration values observed in Chania basin during the investigated time period are presented in Figure 4 and Figure 5, while Figure 6 illustrates the remarkable strong motion intensity values associated with Samos event, despite the great distance of its epicenter. In these figures, the limits noted with a red dashed line on the maps demarcate the old Venetian city, which is a monument of national cultural heritage as well as the most populated part of Chania.

The strong ground motion results from EQ.3 event in Figure 4, show PGA values from 7.5 up to 25 cm/s^2^. The large acceleration values were recorded in the western and central part of the southern Quaternary basin, while the most part within the city limits displayed intermediate values. The PGV projects similar image with the PGA whereas the PGD and Arias intensity exhibit the maximum values in the area between stations PASK, DEYA, and G0LD.

The ground intensity maps for the EQ.4 event in Figure 5, evidence large values of PGA, PGV, and PGD in the central and south eastern part of the research area which is expected as the epicenter is situated south and very close to Chania and there are near-fault directivity effects. Although the PGA values are equivalent to those of the EQ.3 strong event, the rest ground intensity values are considerably lower.

The strong ground motion spatial distribution maps that came out from the SGMN recordings of the Samos strong event occurred in 2020/10/30, are illustrated in Figure 6. Note that station PERI was offline at that period for repair and maintenance reasons. The prominent values of ground shaking intensity are observed in the central part of the sedimentary basin. The PGA values are lower compared to the other two cases, whereas the PGV and PGD values are remarkably higher.

## 5. Discussion and Conclusions

The response spectra from a strong motion sensor after large events provide useful information for the civil protection agencies related to the ground acceleration values which can be compared to the seismic design of buildings curves and assess the potential damage. Having a considerable number of sensors deployed on the different geological formations and city areas with variable vulnerability, makes possible the thorough yet direct examination of the broad urban environment. With the use of several event recordings for a specific station, it is feasible to identify areas with amplified site effects. This outcome can be used both for proactive decisions as well as prompt actions for emergency response and for long-term prevention measures to mitigate earthquake risk, e.g., increase the building stock resilience.

In Chania city, the stations located in loose Quaternary deposits show that there is an increased trend in the spectral acceleration values for the most of studied cases compared to the sensors that are in the stiffer Neogene limestone. The large variation in the acceleration values is due to the considerably different thickness and material stiffness in the Quaternary basin. The available data show that there are site effects from continuous lateral refractions and reflections of trapped waves inside the basin as well as seismic wave reflections on a vertical surface such a suspected fault [12]. An extra focus should be given to these locations as it is very possible to experience more structural damage from a future strong earthquake.

Examining only the PGA spatial distribution maps after a strong ground shaking is not always the best option as the results visualize partial aspects of the total image especially when there are near-fault directivity effects [32]. The maximum acceleration maps between an intermediate depth 6.1 magnitude earthquake with 70-km epicentral distance (EQ.3) and a shallow event with local magnitude 4.5 (EQ.4) and epicenter closer to Chania by 30 km demonstrate similar results. The minimum and maximum PGA values are on the same range while the rest ground intensity map measures for EQ.4 have significant lower values than those for EQ.3.

The strong ground motion sensor networks effectively and promptly provide comprehensive information for a rapid DSS. First, it produces near real time ground intensity parameter spatial distribution maps which provide decision-makers with an image of the potential structural damage. Secondly, by building a database with a series of events, the developed methodology could be incorporated within a rapid DSS for civil protection to simulate or assess long-term seismic-risk reduction options and actions (e.g., land-use, land-use planning, public infrastructure, building codes etc.) as well as short-term ones (e.g., early warning, evacuation plans, emergency actions coordination). It is operational in different spatial and temporal scales and resolutions, visualizes the strong seismic motion effects, and conveys earthquake hazard risk information in a graphical, user-friendly, and intuitive way to support civil protection decision-makers.

## Figures and Tables

**Figure 1 sensors-21-02833-f001:**
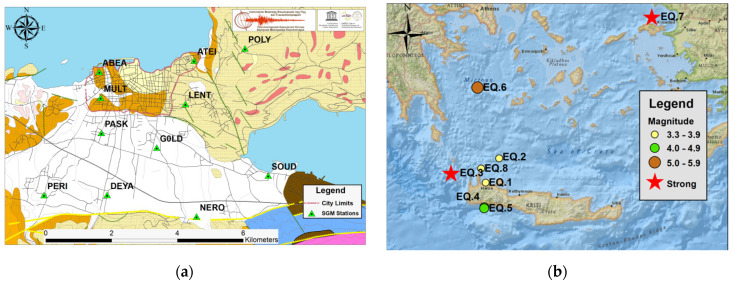
(**a**) The geological setting of Chania metropolitan area and the SGMN topology (see text). (**b**) Epicenters and magnitudes of the studied earthquakes.

**Figure 2 sensors-21-02833-f002:**
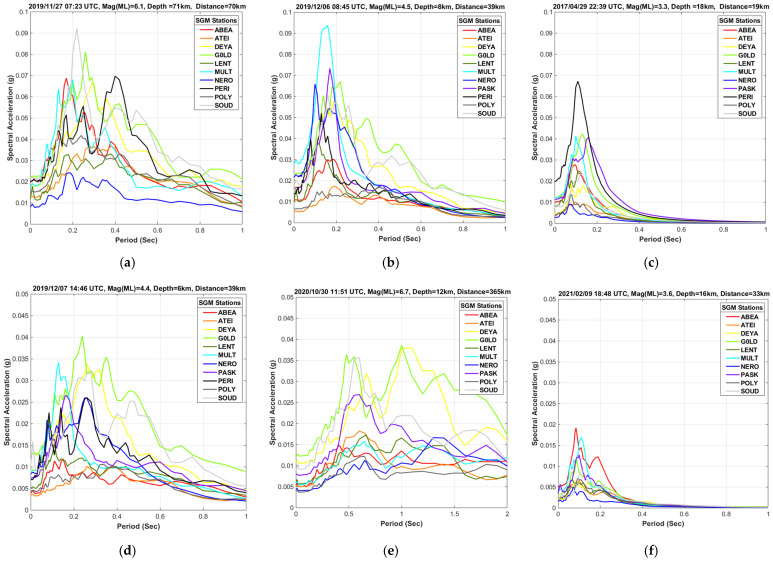
Spectral acceleration in g-units for the strongest events since 2017: (**a**) EQ.3; (**b**) EQ.4; (**c**) EQ.1; (**d**) EQ.5; (**e**) EQ.7 (Samos event); (**f**) EQ.8.

**Figure 3 sensors-21-02833-f003:**
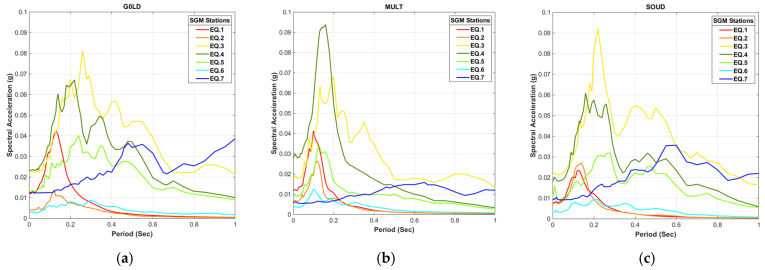
Comparative study of the spectral acceleration values for the same events recorded to selected SGMN stations located on different subsurface geology with contrast of seismic response: (**a**) G0LD-station on Quaternary deposits; (**b**) MULT-station on Neogene marles and Quaternary deposits; (**c**) SOUD-station on loose Quaternary deposits. Stations placed on Neogene limestones: (**d**) ATEI- station; (**e**) LENT-station and (**f**) POLY-station.

**Figure 4 sensors-21-02833-f004:**
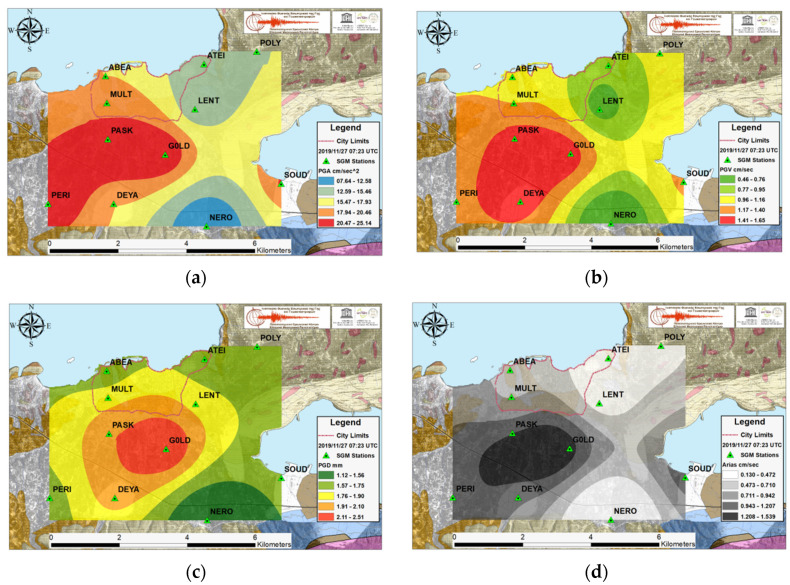
For the EQ.3 strong event, West of Chania, the ground motion parameter spatial distribution results obtained from Ordinary Co-Kriging: (**a**) Peak Ground Acceleration in cm/s^2^; (**b**) Peak Ground Velocity in cm/s; (**c**) Peak Ground Displacement in mm and (**d**) Arias Intensity in cm/s.

**Figure 5 sensors-21-02833-f005:**
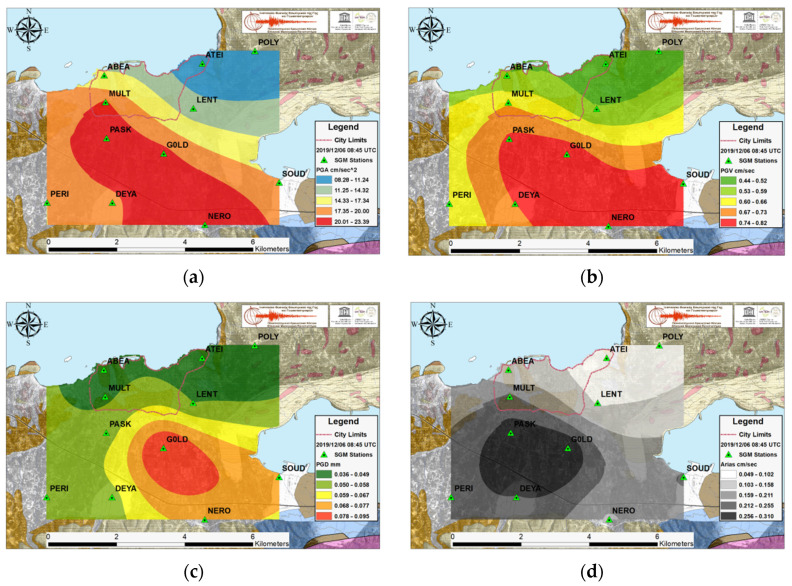
For the EQ.4, South of Chania, the ground motion parameter spatial distribution results obtained from Ordinary Co-Kriging: (**a**) Peak Ground Acceleration in cm/s^2^; (**b**) Peak Ground Velocity in cm/s; (**c**) Peak Ground Displacement in mm and (**d**) Arias Intensity in cm/s.

**Figure 6 sensors-21-02833-f006:**
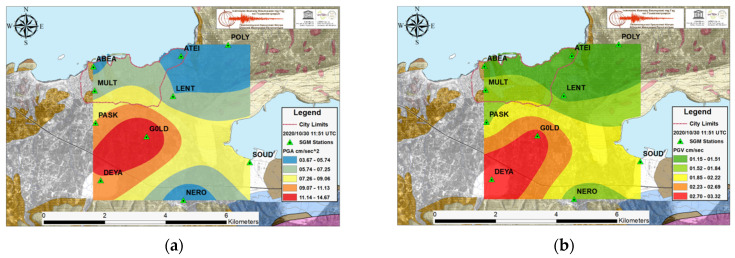
For the EQ.7 Samos strong event, Northeast of Chania, ground motion parameter spatial distribution results obtained from Ordinary Co-Kriging: (**a**) Peak Ground Acceleration in cm/s^2^; (**b**) Peak Ground Velocity in cm/s; (**c**) Peak Ground Displacement in mm and (**d**) Arias Intensity in cm/s. PERI-station was offline.

**Table 1 sensors-21-02833-t001:** Strong ground motion sensors’ location, type and underlying subsoil.

Station	Latitude	Longitude	Sensor	Geology
ABEA	35.5162	24.0113	130SM	Marles with sandstones
ATEI	35.5193	24.0431	130SM	Neogene limestone
DEYA	35.4825	24.0140	ANSS	Quaternary deposits
G0LD	35.4955	24.0306	130SM	Thick Quaternary deposits
LENT	35.5074	24.0403	130SM	Neogene limestone
MULT	35.5091	24.0118	ANSS	Neogene marles-Quaternary deposits
NERO	35.4739	24.0423	130SM	Neogene limestone
PASK	35.4996	24.0121	130SM	Thick Quaternary deposits
PERI	35.4825	23.9928	ANSS	Quaternary deposits
POLY	35.5226	24.0602	130SM	Neogene limestone
SOUD	35.4878	24.0678	130SM	Loose Quaternary deposits

**Table 2 sensors-21-02833-t002:** The earthquake parameters obtained from NOA and the distance form Chania.

ID	Date-Time (yyyy/mm/dd hh:mm)	Latitude (Degrees)	Longitude (Degrees)	Magnitude (M_L_)	Depth (Km)	Distance (Km)
EQ.1	2017/04/29 22:39	35.5472	23.8682	3.3	18	19
EQ.2	2018/01/26 22:09	35.8970	24.1080	3.9	72	44
EQ.3	2019/11/27 07:23	35.6854	23.2599	6.1	71	70
EQ.4	2019/12/06 08:45	35.1906	23.8481	4.5	8	39
EQ.5	2019/12/07 14:46	35.1773	23.8449	4.4	6	39
EQ.6	2020/08/17 07:27	36.9049	23.7250	5.2	90	157
EQ.7	2020/10/30 11:51	37.9001	26.8167	6.7	12	365
EQ.8	2021/02/09 18:48	35.7491	23.7895	3.6	16	33

**Table 3 sensors-21-02833-t003:** The ground motion predominant frequency *f*_0_ and amplitude *A*_0_ at SGMN’s stations.

Station	*f_0_* (Hz)	*A_0_* (Unitless)
ABEA	0.46	2.44
ATEI	0.40	2.27
DEYA	0.44	2.72
G0LD	0.42	4.08
LENT	0.40	2.44
MULT	0.36	2.27
NERO	0.63	1.99
PASK	0.33	3.00
PERI	0.46	3.51
POLY	0.55	2.50
SOUD	0.36	2.26

## Data Availability

The waveforms that have been used for this article are collected from the HC network (https://doi.org/10.7914/SN/HC) (accessed on 2 February 2021) and the waveforms are available through the Greek National EIDA node (http://eida.gein.noa.gr/) (accessed on 2 February 2021).

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
