# Peer review of "Strong Ground Motion Sensor Network for Civil Protection Rapid Decision Support Systems"

_sensors, 2021, doi:10.3390/s21082833_

Round 1
Reviewer 1 Report
The paper focuses on the importance of earthquake hazard risk information. A strong motion sensor network was installed and real-time operated in Chania (Crete Island, Greece) , and strong ground motion data was analyzed based on some typical cases caused by earthquakes in the last couple years. The research is attractive, and the author has done a lot of work. However, a number of critical issues have reduced the readability and quality of this paper. A couple of comments are provided as follows:
1、In Section 2, in terms of data processing methods, the thesis is only expressed in the form of words and language. Can you give some expressions to make the mathematical expression more intuitive?
2、The period of seismic motion survey given in the paper is from 2017 up to February 2021. Will the sensor be damaged or affect the recorded results after working for a long time?
3、In Figure 1 of Section 3, The acceleration results at different positions vary considerably. What is the reason for this? Can the author give an explanation in terms of site effects?
4、In Section 3, the author mentioned: “ … Although, the PGA and PGV values are equivalent with those of the 2019/11/27 event, the PGD and Arias Intensity values are considerably lower.” What is the reason for this result?
5、In Section 3, the author uses a parameter index of Arias Intensity, what is the physical meaning of this index? What are its advantages over peak acceleration?
Author Response
Point-by-point response of the authors to the reviewer’s comments can be found in the uploaded file.

Reviewer 2 Report
Dear authors,
the manuscript (MS hereinafter) “Strong Ground Motion Sensor Network for civil protection rapid Decision Support Systems” by Chatzopoulos et al. submitted for the Special Issue “Sensor Solutions towards Climate-Resilient and Sustainable Cities” of the scientific journal “Sensors” deals with the usefulness and the necessity of a network of strong motion sensors for the management and the reduction of seismic risk in metropolitan areas.
The aims of the MS are completely adequate to the purposes of the special issue, since this study describes the application of a sensor-technology for the sustainability and the resilience of cities characterized by high seismic risk. The MS’s purposes have been achieved by testing the feasibility of the sensor network in the Chania island, southern Greece, by analyzing earthquake and ambient noise data both with traditional and advanced analysis techniques.
The MS is well organized, even if some flaws are present. Here I list the main issues I would like to point out, which refer mainly to the way the manuscript is organized and to some information lacking:
- The introduction well frames, even too much, the social and scientific problem faced in the manuscript. In my opinion, it could be also simplified. Nonetheless, a part describing the state-of-art of the past and present technologies adopted for seismic risk mitigation purposes completely lacks. What kind of networks do seismologist usually use? It is very important, in order to better understand why you are proposing this strong motion sensor network instead of another one.
- Other two questions could arise: Do similar studies carried out in the island exist? Does the suggested methodology allow gaining more insights on the seismic response of different geological formations of Chania with respect to what was already known? A greater emphasis on these aspects would provide originality and novelty to your study.
- It is necessary a distinct section describing the network characteristics and the geological setting: these could not be inserted in the introduction.
- Methodology: More information have to be provided about the signal time window adopted for spectral analysis.
On these grounds, I would suggest a minor revision before the publication of the manuscript.
Here I will provide more specific comments:
Line 43: there is a mismatch between the citation Cioca and Cioca [1] and the reference list, where the first work is Newman
Line 52-57: Too long sentence. Furthermore, what do you mean with mitigation of earthquake hazard?
Line 63-64: “...aiming to improve decision process...”. In my opinion it could be better “...aiming to improve and simplifying decision process...”
Line 80-81: “Hence, seismic-risk reduction planning and prevention strategies should be short-, in-
termediate and long-term monitoring and assessing, considering risk variations.” In my opinion, here you should add a reference to what are the most common worldwide adopted strategies for seismic risk reduction.
Line 94: “possible indirect benefits from earthquake hazard”. What do you exactly mean?
Line 107-128: This part should be moved in one (or two) separate sections.
Line 115-128: It is fundamental to add a geological map showing the location of sensors on different geological formations. In this map, you could also add the epicenter positions of earthquakes adopted in your study (at least the local ones should be shown).
Line 150: “the seismic signal was cut few seconds before P waves arrival”. How many seconds before? What is the length of the time window adopted for the spectral analysis?
Line 186-188: “Secondly, the measured intensity ground motion parameters should remain unchanged as possible”. Unclear concept.
Line 198-207: This paragraph, describing earthquake data, should be moved in the Data section. Furthermore, it is necessary a table summarizing the parameters of the adopted earthquakes. I suggest to number the events, with an ID, starting from 1 to 6.
Figure 1: The same range (Y axis) of Spectral acceleration values should be adopted for different plots, to better highlight differences in seismic response among different stations and due to different earthquake parameters. The same color code describing different stations should be used in different graphs. Furthermore, please increase the font of the x and y label, both titles and ticks.
Figure 2: Two issues: 1) Instead of writing the date and the origin time of the earthquake, it could be better to write the ID of the earthquake, according to the numeration I suggested you to adopt in the earthquake table I proposed you to add. 2) It could be useful to indicate, also with a label, that the top plots refer to stations laying on quaternary deposits, whereas the bottom ones to stations laying on Neogene formations.
Line 270-274: May the NW-SE elongated aspect of PGA and PGD “red zones” be due to the directivity effect of earthquake source?
Line 273: The PGV values are not equivalent to those of the 2019/11/27, as the maximum value is equal to the half of the maximum PGV value obtained for the event of 2019-11-27.
Line 279: Arias Intensity values are higher than the values shown in figure 4.
Line 280: “the PGV and PGD values are remarkably higher” May be due to the predominance of low frequency in the signal? If yes, it has to be clarified.
Author Response
Point-by-point response of the authors to the reviewer’s comments can be found in the uploaded file. Please see the attachment.

Round 2
Reviewer 1 Report
Accept as it is